# Scope of Use and Effectiveness of Dietary Interventions for Improving Health-Related Outcomes in Veterans: A Systematic Review

**DOI:** 10.3390/nu14102094

**Published:** 2022-05-17

**Authors:** Rebecca Mellor, Elise Saunders-Dow, Hannah L. Mayr

**Affiliations:** 1Gallipoli Medical Research Foundation, Greenslopes Private Hospital, Greenslopes, QLD 4021, Australia; saundersdowe@ramsayhealth.com.au; 2Department of Nutrition and Dietetics, Princess Alexandra Hospital, Woolloongabba, QLD 4102, Australia; hannah.mayr@health.qld.gov.au; 3Centre for Functioning and Health Research, Metro South Hospital and Health Service, Buranda, QLD 4102, Australia

**Keywords:** veterans, diet intervention, nutrition, lifestyle, systematic review

## Abstract

Military veterans often have numerous physical and mental health conditions and can face unique challenges to intervention and management. Dietary interventions can improve the outcomes in many health conditions. This study aimed to evaluate the scope of health conditions targeted with dietary interventions and the effectiveness of these interventions for improving health-related outcomes in veterans. A systematic literature review was performed following PRISMA guidelines to identify and evaluate studies related to veterans and dietary interventions. Five electronic databases were searched, identifying 2669 references. Following screening, 35 studies were evaluated, and 18 were related to a US national veteran weight-loss program. The included studies were critically appraised, and the findings were narratively synthesized. Study designs ranged from randomised controlled trials to cohort studies and were predominantly U.S. based. The intervention durations ranged from one to 24 months. The mean subject age ranged from 39.0 to 69.7 years, with often predominantly male participants, and the mean body mass index ranged from 26.4 to 42.9 kg/m^2^. Most dietary interventions for veterans were implemented in populations with overweight/obesity or chronic disease and involved single dietary interventions or dietary components of holistic lifestyle interventions. The most common primary outcome of interest was weight loss. The success of dietary interventions was generally moderate, and barriers included poor compliance, mental health conditions and large drop-out rates. The findings from this review illustrate the need for further refinement of dietary and lifestyle interventions for the management of veterans with chronic health conditions.

## 1. Introduction

Military veterans often experience numerous health issues, such as mental health conditions, insomnia, obesity, chronic diseases and chronic pain [1]. It has been reported that over a third of older veterans have at least three comorbid conditions [2]. The incidence of psychological conditions and risk of physical comorbidities, such as heart disease and hypertension, osteoarthritis, diabetes, chronic pain and lung disease, is greater in the military population than in the general population [3].

The medical management of these complex presentations is challenging. Previous research has evaluated several types of interventions to improve the mental and physical health of military veterans, and these commonly include psychology-based interventions and pharmacological interventions. These interventions have shown mixed results, with some studies showing benefits and others showing limited effects on health in veterans [4,5]. There is great scope for exploring additional approaches to holistic management of the complex individual.

Dietary interventions have been advocated to improve outcomes in numerous health conditions. Various diet interventions have shown to be generally effective (although with some mixed results) for improving outcomes and reducing risk factors for, inter alia, obesity [6], pain conditions [7,8], anxiety and depression [9], insomnia [10], cardiovascular disease [11], diabetes [12] and certain cancers [13]. Concurrently addressing complex health conditions in veterans with dietary management as a component of a multidisciplinary approach appears logical as a safe and beneficial approach and contributes to improving quality of life. 

Attrition rates in various dietary intervention studies can often be high [14], and many different strategies for delivery and implementation have been adopted for dietary interventions in the effort to improve adherence [15]. Poor health behaviours, such as tobacco use, physical inactivity, poor diet and alcohol misuse, are more prevalent in veterans than in civilian populations, and veterans are also more likely to be obese than civilian populations [16]. 

Veterans also tend to have poorer social support [17], which has been linked to physical inactivity and poor chronic-disease management [18] as well as negative physical [19] and mental-health outcomes [20]. These factors may present greater challenges and barriers to lifestyle and behavioural change; hence, the feasibility and effectiveness of implementing dietary interventions in a veteran population may differ from that of other populations with specific health conditions. 

Considering that veterans have a high incidence of poor health behaviours, physical and psychological comorbidities and face unique challenges in undertaking behavioural and lifestyle modifications, it is important to understand how dietary interventions have been utilized and whether they are effective in improving health, specifically in veteran populations. Therefore, the purpose of this systematic review is to evaluate the scope of health conditions targeted with dietary interventions and the effectiveness of these interventions for improving health-related outcomes in veterans. 

## 2. Methods

This systematic review was conducted in accordance with the guidelines of the Preferred Reporting Items for Systematic Reviews and Meta-Analyses (PRISMA) statement [21] (Appendix A) and was registered on PROSPERO (International Prospective Register of Systematic Reviews, No CRD42021236259).

### 2.1. Literature Search

We searched the electronic databases MEDLINE (via PubMed), CINAHL (via EBSCO), Embase, Cochrane CENTRAL and PsycINFO (via Ovid) from inception to 23rd February 2021, using combinations of MeSH and free-text words for “veteran” and “diet” (Appendix A). The search was originally designed in PubMed, and then the selected terms and their synonyms were translated for the respective databases using Polyglot [22].

### 2.2. Study Eligibility Criteria

The study inclusion criteria were developed using the Participants, Intervention, Comparator and Outcome (PICO) [23] plus Study design approach (Table 1). Participants were veterans of military or defence forces from any nation or discipline. Interventions were dietary interventions conducted at any time post-discharge from the military or defence force. Dietary interventions could include specific dietary patterns, altered specific nutrients or foods through dietary intake and/or energy intake adjustments. 

Studies were eligible regardless of the mode of intervention delivery or intervention duration and whether diet was part of a multi-factorial intervention with other lifestyle or behavioural components (e.g., diet and exercise or psychological support). A control or comparator group was not required, and hence single-group pre–post intervention studies and randomised or non-randomised controlled trials involving a usual-care, no-care or alternative-intervention group were eligible. 

The outcomes of interest were any health-related outcome measures, including anthropometric or body composition, dietary intake or behaviours, cardiometabolic risk markers, quality of life, mental health, physical function or strength, patient-reported outcome measures and chronic disease incidence or endpoints. All study designs that involved a dietary intervention were included. This included cohort studies if they followed up and evaluated participants undertaking a diet intervention that was considered to be usual care. Studies were required to be published in English for inclusion, and no publication date restrictions were imposed. 

### 2.3. Screening and Data Extraction

Identified references were imported into Endnote X9 reference management software [24], and de-duplication was conducted using the Endnote duplication tool. The de-duplicated set was imported into Covidence [25]. Two reviewers (HM, RM or ESD) independently screened titles and abstracts with disagreements resolved by consensus. The full text of articles that were considered potentially eligible were then screened for eligibility independently by two reviewers (HM, RM or ESD) and agreement was reached via group consensus. Data were extracted from each study into pre-defined tables by one reviewer and cross-checked by a second reviewer (RM, ESD or BS performed either step) and included the study citation, study design, participant eligibility and characteristics, intervention details, control details if relevant, outcomes of interest and results.

### 2.4. Risk of Bias Assessment

Risk of bias in each included study was assessed independently by two reviewers (RM, ESD or BS) using the Academy of Nutrition and Dietetics Quality Criteria Checklist: Primary Research [26]. Studies were assessed on 10 key criteria assessing the internal and external validity. These criteria related to the clarity of the research question, selection bias, similarity of study groups, methods of handling withdrawals, blinding procedures, intervention details; measurement, validity and reliability of outcomes, statistical analyses methods, appropriateness of conclusions and bias related to funding or sponsorship. Each study was awarded an overall positive, neutral, or negative method quality rating based on the scoring tool instructions. Papers were not excluded based on the quality criteria. 

### 2.5. Data Synthesis

Due to the fact that the characteristics of the studies (the different study designs, the large range of different outcomes measured, the use of different types of dietary interventions and the different health conditions targeted) were too diverse to yield a meaningful summary estimate of effects, meta-analyses were not deemed suitable. Instead, we provide a narrative synthesis of the findings, structured around the types of intervention (i.e., specific dietary regime, dietary education and behavioural change), target population characteristics (e.g., health condition) and type of outcome (e.g., weight change, quality of life and metabolic indices). 

## 3. Results

### 3.1. Study Selection

The literature search identified 2669 references after the removal of duplicates, and 2548 were removed after title and abstract screening. Of 121 references retrieved for full text screening, 86 were excluded. The remaining 35 studies were included in this review (Figure 1). Eighteen of these studies were related to a U.S. national program, MOVE! (Managing Overweight/Obesity for Veterans Everywhere), which is an ongoing clinical weight management program for veterans and these are discussed separately. 

### 3.2. Risk of Bias 

The results of the quality assessment criteria are presented in detail in Table 2. In summary, 12 studies received a neutral rating and were of moderate methodological quality, while the remaining 23 studies were of high methodological quality, receiving a positive rating. No studies received a negative rating. The main source of bias was a lack of blinding of participants and investigators to the treatment group, as well as of data collectors to the outcome measures.

### 3.3. Study Characteristics

The characteristics of the 17 studies not associated with the MOVE! program are described in Table 3. Most studies were conducted in the USA (*n* = 15 of 17), and there was one each from Australia and Taiwan. Of these, ten were randomised controlled trials (RCTs) [27,28,29,30,31,32,33,34,35], three were dietary interventions in usual care that were evaluated using routinely collected data accessed retrospectively [36,37,38], three were single arm interventions [39,40,41], and one was a cohort study [42]. The trial intervention duration ranged from 1 to 24 months. The mean age of the study participants ranged from 39.0 ± 6.7 to 69.7 ± 0.7 years. Most studies had a predominance of male participants, with the percentages of males ranging from 42% to 100%. The mean body mass index (BMI) of participants ranged from 26.4 ± 2.6 to 42.9 ± 7.7 kg/m^2^.

### 3.4. Veteran Populations and Dietary Interventions

In studies of overweight or obese veterans (*n* = 8), commonly the primary aim was to achieve weight loss. Three studies investigated a low-carbohydrate diet compared to either a low-fat/low-calorie diet [33,43] or a low-fat diet combined with Orlistat therapy [35]. One study compared an intermittent energy restriction diet plan (the 5:2 diet) with a standard energy restricted diet [29]. One trialled individualized wellness coaching, which addressed healthy eating habits, shopping and cooking advice and used the stage-of-change model to alter eating behaviours [32]. 

One examined the feasibility and efficacy of weekly educational mailings and telephone consultations that addressed weight management issues in reducing weight and improving dietary habits [28]. Two studies included overweight veterans with schizophrenia: one was an inpatient regime consisting of a calorie-restricted diet and physical exercise [34], whilst the other evaluated a psychosocial weight management program that focused on tailored nutritional and behavioural change education [39]. 

Three studies were conducted in veterans with type II diabetes. One study compared a low-carbohydrate diet to a low-fat diet [31]. Another study incorporated dietitian-led sessions in a group education programme with a focus on carbohydrate intake, dietary fats and general healthy eating, including for weight loss [37]. The last study evaluated The Healthy Teaching Kitchen programme, which comprised cooking and nutrition education classes with topics on carbohydrate counting and meal planning [36]. 

All other study populations varied. One study in veterans with uncontrolled hypertension compared a telephone-delivered intervention of monthly counselling for exercise, diet and medications based on the current stage of change with sessions of non-tailored information and usual care [30]. As part of a preliminary phase of a larger trial, patients with primary hypercholesterolaemia received two to four dietitian intervention sessions over 6–8 weeks focused on reducing the intake of total and saturated fat, cholesterol and energy [38]. A ketogenic diet was implemented in a population of veterans with advanced malignant cancers [41] to evaluate safety and tolerability. 

In a population of veterans with Gulf War Illness (GWI), a low-glutamate diet was implemented to examine its effectiveness on symptomatology [42]. A study in participants with amnestic mild cognitive impairment compared a low-saturated-fat/low-glycaemic-index diet to a high-saturated-fat/high-glycaemic-index diet [27]. Finally, a study in nursing home residents reported the feasibility of implementing a dietary intervention that addressed age-related nutrition topics, such as hydration, meal planning, the Dietary Approaches to Stop Hypertension (DASH) diet and protein intake, as part of an exercise- and health-promotion programme [40].

### 3.5. Attendance and Attrition

A number of interventions utilized group sessions and reported the number of sessions attended (Table 3). Shorter or less-intense group interventions appeared to have better attendance rates [36,37]. However, in the program for nursing home residents where attendance was optional, a mean of only 2.9 of 7 available classes were attended [40], and mean attendance at 16 offered weekly sessions in the psychosocial weight management program for obese veterans with schizophrenia was poor at 3.8 [39]. Longer duration studies that reassessed participants at 48 weeks reported reasonable percentages of participants that attended for final follow-up measures, ranging from 79% [35] to 88% [44]. In a study offering group sessions over 24 months, only 47.22% completed the 24 month assessment [31]. 

Other studies utilized individual sessions to deliver the intervention. Number of attended sessions was only reported in one study, in which participants attended a mean of 2.8 of the required two to four dietitian consultations over 6 to 8 weeks [38]. Two 6-month RCTs reported that 65% [32] to 75% [29] of participants in the intervention groups completed the studies. An RCT that provided weekly counselling sessions for four weeks, then monthly sessions for 11 months [33] reported that 60% attended the 6-month follow up appointment, and 67% attended the 12-month follow up appointment.

Some studies delivered intervention content via phone calls. One study reported that 87% of the intervention group, which received weekly mailings and phone calls, and 55% of the comparison group, which received “usual care” from a hospital clinic, attended 8-week follow-up [28]. Another study using dietary training via Skype reported that 87% completed the trial, and 74% completed the 3-month follow up [42]. Of these, 88% were still following the diet. 

In one study in which the intervention was strongly controlled, with food delivered to the homes of participants twice weekly, all participants completed the trial [27], whilst in another study in which inpatient participants were provided with a diet overseen by the hospital dietitian for six months [34], all participants who remained in hospital completed the trial. In a trial to evaluate the safety and tolerability of a ketogenic diet in patients with advanced malignant cancer, 64.7% completed 4–16 weeks of dieting, and at 16 weeks, only 36% (four participants) still maintained the diet [41]. 

### 3.6. Health-Related Outcome Measures

Data and statistical outcomes for all study results are reported in detail in Table 4, with the main results for health-related outcomes, as described by the individual studies, summarised in the following text.

### 3.7. Weight and BMI 

We found that 11 of 17 studies were interested in weight change as their primary outcome, reporting either total weight loss, weight loss percentage, or percentage of participants that lost ≥5% of baseline body weight. Of these, seven studies reported a statistically significant reduction in weight post-intervention (range of mean change: −11.4 kg to −0.2 kg) [29,32,34,35,36,37,44]. Four studies also included BMI as a primary outcome, and three reported a statistically significant reduction in BMI with diet intervention (range of mean change: −1.5 to −0.35 kg/m^2^) [32,34,36]. Only two of these studies reported a significant between-group difference in weight and/or BMI change at the final follow-up [34,37]. 

### 3.8. Blood Pressure

There were 2 of 17 studies reported change in blood pressure (BP) as a primary outcome [30,37]. Of these, only one study, comparing a diet intervention tailored to stage of change and a non-tailored health education intervention with usual care, reported a significant reduction in BP in both intervention groups at six months, as well as a significantly greater number of participants with controlled BP in the tailored group compared with other groups [30]. 

### 3.9. Blood Composition or Metabolic Parameters 

Haemoglobin A1c (HbA1c), glucose, insulin and serum lipids were often included as primary outcome measures. A basic diabetes education intervention compared with standard diabetes management significantly reduced HbA1c levels in veterans with type II diabetes [37], whilst a Healthy Teaching Kitchen program reported no significant change [36]. A calorie-restricted diet for hospitalized obese veterans with schizophrenia produced a significant reduction in insulin levels over the six month intervention, although there were no significant differences in glucose or insulin levels compared with the control group [34]. 

Three studies assessed serum total cholesterol (TC), low-density lipoprotein cholesterol (LDL-C), high-density lipoprotein cholesterol (HDL-C) and triglyceride levels. There was a significant reduction in triglyceride levels after therapy focused on reduction in saturated fat, cholesterol and energy intake [38] in veterans with primary hypercholesterolaemia and an inpatient calorie-restricted diet and exercise regime in obese veterans with schizophrenia [34]. However, only Sikand and colleagues reported a significant decrease in TC, LDL and HDL [38]. There were no significant changes in serum lipids post-intervention in the Healthy Teaching Kitchen intervention [36]. 

### 3.10. Other Health-Related Outcomes

One study examined fruit and vegetable intake as the primary outcome of interest and found no significant change in daily servings in older veterans after participating in a dietary education component within an exercise and health promotion program [40]. The total symptom score (total number of typical GWI symptoms experienced) was significantly decreased in veterans with GWI compared to a wait-listed control group after following a low glutamate diet [42]. 

The safety and feasibility of a ketogenic diet for participants with advanced cancers examined measures, such as adverse effects, weight, BMI, BP, haematology, ketones, lipids, glucose/ketone indices, quality of life parameters and effects on the tumour. This trial demonstrated that the ketogenic diet was well tolerated by those who remained on the diet (the reasons for drop-outs were not dietary related), and adverse effects were minimal in those remaining on the diet throughout the observation period [41].

### 3.11. The MOVE! Weight Management Program

In the US, the Veterans Health Administration (VHA) identified a need for effective weight management interventions for their veterans, which prompted the development of the MOVE! (Managing Overweight/Obesity for Veterans Everywhere) Weight-Management Program. Details about the current MOVE! program are reported elsewhere (www.move.va.gov, accessed on 9 March 2021). The MOVE! program has evolved over time to include TeleMOVE (i.e., delivering the MOVE! program via videoconferencing technology), MOVE! plus adjunctive treatments, as well as variance in the length of program and content advancement. 

Numerous studies have been conducted, including evaluative, feasibility and comparative studies, to ensure that the MOVE! programs and other weight loss opportunities continue to be developed and refined. The database search for this review identified 18 studies related to the MOVE! program. The study characteristics, data and outcomes for all MOVE! study results are reported in detail in the Appendix A with a very brief overview reported in the following text.

### 3.12. Health-Related Study Outcomes 

Weight change was the primary outcome measure in the majority of the MOVE!-related studies. Two trajectory studies reported that veterans had been progressively gaining weight prior to enrolment in the MOVE! program and, once enrolled, began to lose weight at a steady rate, ranging from an average of −1.6 kg/year [45] to −2.2 kg/year in the first year of participation [46], depending on the level of involvement. 

Mental health had an impact on effectiveness of the program, as those with mental health diagnoses were not as successful in losing weight [47], despite modifications and adaptations to the program for those with serious mental illness [48,49]. However, a short pilot study (over 10 weeks) showed that the MOVE! program was able to reduce the severity of depression in obese veterans with severe depression to a similar extent as a 2-week intense residential-based program [50]. 

The introduction of telemedicine (e.g., TeleMOVE!) was demonstrated to be a successful method of delivery, as all related studies reported significant weight loss for the intervention group (despite different study comparators and study durations) ranging from −3.9 to −5.3 kg [51,52,53]. The addition of personal digital assistants to self-monitor diet and physical activity and biweekly coaching calls was also successful in optimising weight loss [54]. 

However, the concurrent treatment of apathy with pharmacotherapy was no more effective than the standard MOVE! program in achieving weight loss [55]. The introduction of a nutrigenetic-guided diet into the MOVE! program was no better than a standard balanced diet in its ability to achieve a loss of ≥5% of body weight in participants [56]. Improved healthy eating habits, in terms of a greater intake of fruit and vegetables, was enhanced by the introduction of tailored newsletters [57].

When compared to two other national weight-loss programs, MOVE! was not inferior over the longer term. When compared to the Aspiring for Lifelong Health (ASPIRE) program, participants in an ASPIRE-Group arm lost more weight than an ASPIRE-phone group and MOVE! participants over 12 months [58]. However, by 24 months, none of the three programs proved superior in weight-loss success [59]. In addition, in obese veterans with type II diabetes, those participating in the Veterans Affairs Diabetes Prevention Program (VA-DPP) had lost more weight than the MOVE! group by six months; however, by 12 months, there was no significant difference in weight loss between the groups [60]. 

Promisingly, it has been found that MOVE! participation was associated with a reduced incidence of total cardiovascular disease, coronary artery disease, cerebrovascular disease, peripheral vascular disease and heart failure over a follow-up period of almost five years [61]. 

## 4. Discussion

This systematic review aimed to explore the scope of use and effectiveness of dietary interventions for improving health-related outcomes in veterans. The majority of health conditions that were addressed by dietary interventions were chronic diseases or illnesses, such as obesity, type II diabetes, hypercholesterolaemia, uncontrolled hypertension and advanced malignant cancers. Psychological conditions were also targeted, such as amnestic mild cognitive impairment, schizophrenia and the symptoms of GWI. 

This range of conditions targeted with dietary interventions is unsurprising. In the United States, 35% of the Veterans’ Health Administration (VHA) primary care enrollees (which represents 90% of all of VHA patients) are estimated to be obese [62], which puts them at a higher risk for chronic diseases, such as hypertension, dyslipidaemia, stroke, diabetes, coronary heart disease and osteoarthritis as well as various forms of cancer [63]. 

Additionally, the prevalence of mental-health disorders and alcohol-use disorders is higher in veterans than in the civilian population [64], and post-traumatic stress disorder (PTSD) is associated with greater BMI through depression–that is, higher symptoms of depression are associated with poor lifestyle behaviours, such as less physical activity, poorer diet and a greater likelihood of smoking in veterans [65]. All of these conditions can potentially benefit from dietary interventions and lifestyle changes. 

Dietary interventions were either delivered as a specific form of diet or incorporated as part of a holistic lifestyle management approach. The range of specific diet regimes evaluated included low-carbohydrate diets, low-fat diets, ketogenic diets, continuous energy restricted diets and the 5:2 diet. One study even examined the feasibility of implementing a nutrigenetic-guided diet into individuals’ management in order to improve weight-loss outcomes [56]. 

Notably, most of the dietary components of interventions were focused on nutrients or restriction-based recommendations rather than healthy dietary patterns or overall diet quality. Dietary guidelines for prevention and management of chronic disease have evolved such that overall eating patterns that focus on whole foods and their combinations, rather than isolated nutrients, such as Mediterranean and DASH dietary patterns, are recommended [12,66,67,68,69]. The current review suggests there is a gap in the literature investigating the effect of dietary patterns on health outcomes in veteran populations.

Holistic weight management programs address many lifestyle factors (diet, exercise, psychological factors and interaction/socialization), and the dietary components of these types of programs place emphasis on education and strategies to facilitate behaviour modification. Obesity is considered to be a chronic disease with multifactorial aetiology, and thus addressing modifiable lifestyle factors (such as proper nutrition, regular physical activity and attention to eating behaviours) is key to success. Even moderate weight loss can lower the risk of other obesity-related comorbidities [70].

A number of studies implemented a stage-of-change model, which recognises that people move through a series of stages when adopting a new behaviour and that different treatment approaches and health communication strategies may be necessary for individuals in the different stages of change [71]. Research across numerous health conditions has shown improvements in recruitment, retention and progress using stage-matched interventions, with promising outcomes also found with computer-based individualised and interactive interventions [72]. 

This strategy appeared to be successful in reducing and controlling BP in veterans with hypertension and uncontrolled BP [30], as well as improving dietary intake and reducing weight in a population of obese veterans [32]. However, both of these studies were performed over six months, and it remains to be seen if these behavioural and dietary changes are sustained over the longer term. 

Adults with serious mental illness (SMI) are more likely to be overweight or obese, which contributes to a greater risk of comorbid medical conditions, such as type II diabetes and cardiovascular disease [73]. Lifestyle interventions in people with SMI show moderate promise but are not always successful [74]. People with SMI may also have cognitive deficits, limited literacy and challenging social situations, which can affect the success of interventions to address their health conditions. For example, one study found that participants with PTSD and other mental health conditions who were enrolled in the MOVE! program lost less weight than those with no mental health conditions [47]. 

However, attempts to improve outcomes by adapting the methods of delivery of the MOVE! program appear disappointing. Strategies, such as concurrent treatment for apathy [55], made no additional difference to weight loss. A modified, tailored MOVE! program for veterans with schizophrenia failed to improve uptake—only 56% of participants completed the six month assessment, only seven participants lost 5% of their body weight [48], and there were no differences in weight loss, metabolic, dietary, physical activity, attitudinal or functional measures from a control group who received only basic lifestyle information. 

Delivery of MOVE! to obese veterans with SMI via internet browser-based educational modules, self-tracking activity and weight, individualized homework and weekly telephone calls from peer coaches (WebMOVE!) also failed to improve outcomes, as these participants only completed 49% of the modules, and an in-person MOVE! group only completed 41% of the modules. A sub-analysis of only the obese participants reported that merely 26% lost 5% of their body weight at six months [49]. These studies emphasize the need for continued efforts to customise and adapt lifestyle and health interventions to improve uptake and adherence in veterans with mental-health conditions. 

Often isolation, distance from healthcare facilities, long travel times, inconvenience and expense are substantial barriers to attending face-to-face weight management programs or accessing healthcare professionals. Of the studies that examined whether videoconferencing, telemedicine, phone counselling/coaching and mail-outs were effective methods of delivering weight management programs to veterans who face these difficulties, all reported successful weight loss with the intervention, and of the five studies with a comparator group/s, two reported significantly more weight loss with the intervention. This is an encouraging finding and a positive step towards using telemedicine and phone-based methods to expand and facilitate delivery of these services to veterans where accessibility and remoteness may otherwise preclude them from valuable healthcare. 

Apart from two trajectory studies conducted over four and eight years [45,46] the duration of the remaining studies ranged from one month to 24 months. As the majority of studies were focused on weight loss, limitations to the interpretation of results, including inadequate study duration, large proportions of subjects lost to follow-up, or a lack of appropriate usual care or control group, should be recognised. Many of the studies on specific diet regimes were of shorter duration, which may not allow for significant weight loss, or reflect the continuation of healthy eating habits and sustained weight management. Generally, nutrition education interventions that last for longer than five months report a higher level of success, particularly as behavioural changes take time and practice [75]. It has been reported that dietary and lifestyle therapy generally provides <5 kg of weight loss after 2–4 years [76]. 

Adherence, engagement and compliance with intervention protocols is vital to improving health-related outcomes. Not all studies reported attendance rates at group or face-to-face interventions or adherence to dietary regimes. A number of studies reported the proportions of how many participants attended final follow-up sessions but not whether participants actually completed or adhered to the intervention. Adequate adherence is one of the difficulties encountered in weight loss and dietary studies and is an aspect that must be considered when interpreting the results from these studies. In general, attendance was better for shorter-term group sessions. 

However, in longer-term group session programs, adherence was not particularly high. The importance of engagement on successful outcomes was emphasized by Yancy and colleagues, where participants were given their preferred choice of diet. Those in the intervention group attended an average of 13.5 of 19 attendances (71% of those possible). However, in a sub-analysis, it was shown that participants who attended at least 15 group sessions had a greater mean weight loss than those who attended fewer than 15 sessions [44]. Some identified barriers to maintaining participant engagement in dietary intervention studies include disinterest in the study, difficulty attending sessions or frustration with a lack of weight loss [31]. 

The majority of studies were performed in overweight or obese veterans, or those with type II diabetes, and most of these reported improvements in body weight, BMI, waist circumference, as well as physiological biomarkers. This is promising in terms of improved health-related outcomes, as it has been shown that, in overweight individuals with impaired glucose tolerance, for every kilogram of weight lost, there is a 16% reduction in the risk for progression to type II diabetes [77]. In type II diabetes, improvement in fasting glucose, HbA1c, triglycerides and systolic BP may begin at only 2–5 kg of weight loss, with greater weight loss producing greater benefits, while improvements in diastolic BP and HDL cholesterol may be seen at 5–10 kg of weight loss [78].

Although most studies reported significant reductions in the baseline weight or BMI post intervention, fewer studies reported significant differences between the intervention group and comparator or control groups post intervention, as both groups lost significant amounts of weight. In the studies that compared different dietary regimes, none were able to demonstrate superiority in weight loss potential between the diets, and only two of these studies noted improvements in blood pressure, triglycerides or HbA1c in obese participants following low calorie diets, many of whom also had type II diabetes. 

Although the majority of papers focused on weight loss as their primary outcome measure, dietary interventions may also have a positive impact on other health outcomes, such as mental health and health-related quality of life, particularly in populations with multiple comorbidities. Consuming a wide variety of healthy foods, including fruits, vegetables, whole grains, legumes and fish containing omega-3 fatty acids, can have positive effects on mood [79] and levels of depression [80], and the quality of life can be enriched by improved physical functioning, tolerance to pain, general health perceptions and vitality and can lead to fewer role limitations due to physical health problems [81,82].

### Strengths and Limitations 

The strengths of this study include adherence to a systematic methodology and use of the PRISMA guidelines. However, a number of limitations must be acknowledged in this review. It is widely recognized that randomized controlled trials are the gold standard in terms of study design and provide the highest quality scientific evidence. Non-randomized studies are inherently at risk of bias. However, it is also acknowledged that, in order to accommodate the evaluation of various research questions (such as the effectiveness or efficacy and outcomes, such as survival or severe adverse events), the inclusion of more than one study design appears to be necessary in systematic reviews of healthcare interventions [83]. 

In order to address the aim of our research question, which was to evaluate the scope of health conditions targeted with dietary interventions and the effectiveness of these interventions for improving health-related outcomes in veterans, it was necessary to design our search strategy to identify papers with numerous study designs involving interventions. This resulted in heterogeneous populations, different outcome measures and often small populations. Subsequently, meta-analysis of the results was not suitable. 

It is acknowledged that the inclusion of dietary interventions that were part of multifactorial lifestyle or behavioural interventions in our search criteria does not allow for discrimination of the effects of diet from effects due to the other lifestyle components in the program (e.g., exercise or psychological approaches). However, this is commonly how numerous medical and psychological conditions are addressed, and it aligns with clinical guidelines. The inclusion of diet-only and diet as part of combined interventions allowed exploration of the full scope of dietary interventions that have been investigated across specific veteran populations and to include pragmatic studies. 

This search strategy identified 18 studies related to the MOVE! program. Although available to all eligible US veterans, participation and attendance at sessions is voluntary. Therefore, a major selection bias is evident in these studies as well as being limited by a high drop-out rate. Most of the MOVE! studies were focused on the evaluation of program success or the effectiveness of adaptations for sub-populations (e.g., mental health conditions) and commonly used retrospective chart reviews in order to achieve this. This often resulted in a degree of missing data. In acknowledgement of these differences, we reported the MOVE! results separately from those of the other identified papers. 

Although there were no overall negative ratings for the quality of the studies, a number of biases can be identified. Although lack of blinding was almost inherent, methods of handling withdrawals was a common source of bias, largely due to inadequate descriptions of the number and characteristics of the withdrawals or whether all enrolled subjects were accounted for. 

A number of studies with small subject numbers were included, impacting the power of the results and analyses. The majority of studies included in this systematic review were US-based and predominantly involved male veterans; thus, the findings are not able to be generalised to the wider international veteran population, nor to female veterans. Additionally, few papers reported longer term follow-up assessments to determine whether the behavioural and lifestyle changes achieved in the initial relevant interventions were maintained over time–an important component in lifelong health and wellbeing.

An overview of these studies provide a sense of the difficulties and challenges faced during dietary research and when attempting to implement dietary interventions to veterans with various physical and psychological conditions. Although generally successful to some degree in achieving the primary aims of the studies, such as weight loss or metabolic/physiologic improvement, dietary studies must continue to strive for better methods to improve attendance and adherence rates. 

Many of these studies had small subject numbers, and thus larger studies need to be performed to strengthen the validity of the results and the quality of the evidence. Strategies to address psychological issues (such as PTSD, depression, SMI, apathy and disinterest) need to be further developed in conjunction with dietary interventions, particularly for veterans where these issues are prevalent. Additional longer term studies ought to be performed in order to assess the sustainability of the interventions and their effect on lifelong health and wellbeing, as behavioural change takes time. 

It could be suggested that more successful outcomes might be achieved by considering use of a stage-of-change model to improve recruitment, retention and progress and by adopting implementation strategies seen to be of benefit, such as computer-based individualized and interactive interventions. Promoting the use of healthy diet patterns in conjunction with lifestyle modifications and addressing social and psychological issues not only aligns with dietary guidelines for the prevention and management of chronic disease but also contributes to improvements in lifelong health, wellbeing and quality of life.

## 5. Conclusions

This review identified a large range of conditions in military veterans that have been targeted by dietary interventions with only moderate success. Most commonly, obesity and associated comorbid chronic conditions were addressed with either single dietary regimes or holistic lifestyle management programs with diet addressed as a component. Limited studies targeted an overall healthy dietary pattern. Barriers to success included poor attendance and adherence rates, mental health conditions and often large drop-out rates. However, the findings from this review illustrate the need for further refinement and the development of dietary and lifestyle interventions for the management of veterans with chronic health conditions. 

## Figures and Tables

**Figure 1 nutrients-14-02094-f001:**
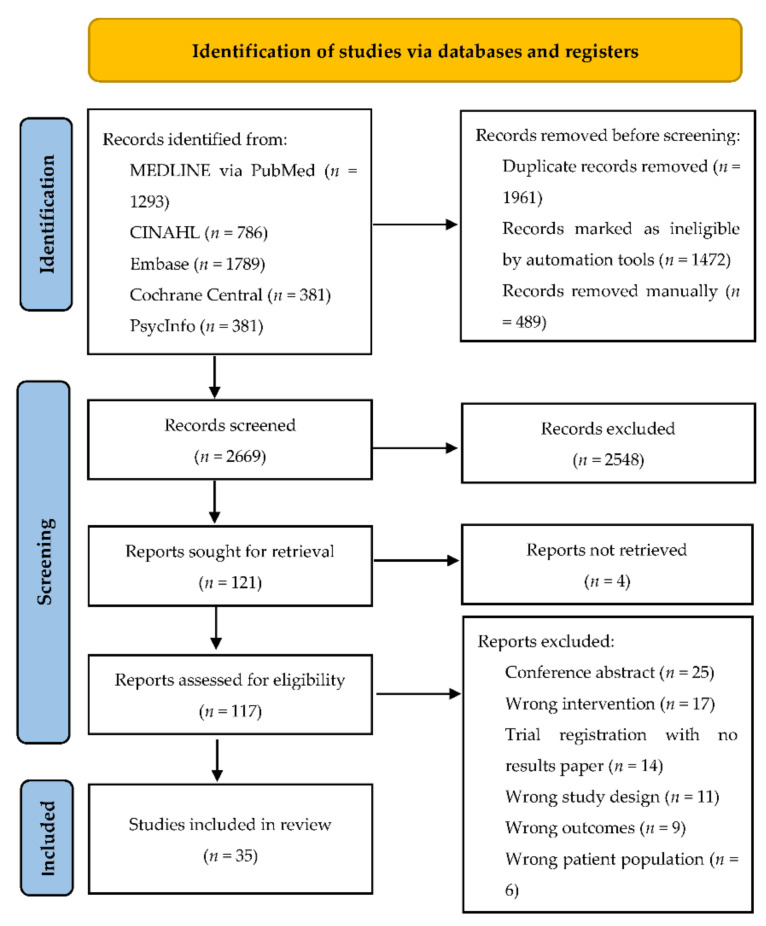
PRISMA flow diagram for our systematic search.

**Table 1 nutrients-14-02094-t001:** PICOS study eligibility criteria.

Criteria	Inclusion	Exclusion
P (Participants or population)	veterans of military or defence forces any nation any discipline (army, navy or air force) any health condition (psychological or physical) targeted for prevention or treatment	if the population includes participants who are not veterans (unless relevant outcomes are reported on separately for veterans)
I (Intervention)	dietary intervention (may include counselling or education or targeting the environment) any mode of delivery (i.e., individual or group, face to face or telehealth) any duration may include/be classified as: (a) altered overall diet (e.g., prescribed a specific dietary pattern, such as vegetarian, Mediterranean or population-based dietary guidelines) (b) altered specific nutrient/s or food/s through dietary intake and/or (c) energy intake adjustment (e.g., continuous energy restriction or fasting protocols) conducted any time post-discharge from the military or defence force diet may be part of a multi-factorial intervention that includes other lifestyle or behavioural components (e.g., exercise or psychological support)	interventions where the only dietary-related component is a nutraceutical or supplement without other changes to intake of foods or nutrients if the dietary intervention is part of a multi-factorial intervention that includes pharmacological or medical therapies
C (Control or Comparator)	any or none control or comparator groups may receive usual or no care, or an alternative intervention	
O (Outcomes)	any health-related outcome measures that are relevant to diet may include the following measures: ◦ anthropometric or body composition (e.g., weight, BMI, waist circumference and body fat percentage) ◦ dietary intake or behaviours (e.g., dietary scores, nutrient or food intake and eating habits) ◦ cardiometabolic risk markers (e.g., blood lipids, blood pressure and inflammation) ◦ quality of life or mental health ◦ physical function, fitness or strength ◦ fatigue ◦ pain ◦ other patient-reported outcome measures ◦ chronic disease incidence or endpoints	studies that report on qualitative outcomes only
S (Study design)	all studies involving a dietary intervention (randomised controlled trials, non-randomised controlled trials, pseudo-randomised controlled trials, single arm pre-test/post-test studies and cohort studies that involved follow up and evaluation of a usual care intervention)	case studies, letters, editorial, reviews or conference abstracts (where the study of the conference abstract has not been published in a full article)

**Table 2 nutrients-14-02094-t002:** Risk of bias outcomes.

	Bayer-Carter, 2011	Boutelle, 2005	Conley, 2018	Dexter, 2019	Friedberg, 2015	Holton, 2020	Iqbal, 2010	Niv, 2014	North, 2015	Serra, 2021	Shahnazari, 2013	Shiroma, 2020	Sikand, 1998	Stern, 2004	Tan-Shalaby, 2016	Wu, 2007	Yancy Jr, 2010	Yancy Jr, 2015	**MOVE!**	Ahrendt, 2014	Allicock, 2010	Dahn, 2011	Damschroder, 2010	Damschroder, 2014	Desouza, 2012	Frankwich, 2015	Goldberg, 2013	Hoerster, 2014	Jackson, 2017	Lutes, 2017	Moin, 2017	Romanova, 2013	Rutledge, 2017	Skoyen, 2015	Spring, 2013	Young, 2017
**Overall Quality Rating**	+	+	+	+	+	□	+	□	□	□	+	+	□	+	+	□	+	+		+	□	□	+	+	□	□	+	+	□	+	+	+	+	+	+	□
Relevance Questions^a^																																				
All studies received a Yes rating for all Relevance questions.																																				
Validity Questions																																				
1. Was the research question clearly stated?																																				
2. Was selection of study subjects free from bias?																																				
3. Were study groups comparable?																																				
4. Was method of handling withdrawals described?																																				
5. Was blinding used to prevent introduction of bias?																																				
6. Were interventions and any comparison(s) described in detail?																																				
7. Were outcomes clearly defined and measurements valid and reliable?																																				
8. Was the statistical analysis appropriate for the study design and type of outcome indicators?																																				
9. Are conclusions supported by results with biases and limitations taken into consideration?																																				
10. Is bias due to study’s funding or sponsorship unlikely?																																				

Overall Quality Rating: + = Positive, **□** = Neutral, - = Negative; Green= Yes, Yellow = Unclear, Red = No, and Light grey = Not applicable. (NB—MOVE! studies separated from other studies by dark grey central line). ^a^ Relevance questions: (1) Would implementing the studied intervention or procedure (if found successful) result in improved outcomes for the patients/clients/population group? (Not applicable for some epidemiological studies); (2) Did the authors study an outcome (dependent variable) or topic that the patients/clients/population group would care about?; (3) Is the focus of the intervention or procedure (independent variable) or topic of study a common issue of concern to dietetics practice?; (4) Is the intervention or procedure feasible? (NA for some epidemiological studies).

**Table 3 nutrients-14-02094-t003:** Characteristics of the included studies, grouped and listed in order of health condition discussed in text.

Study, Location and Design	Health Condition	Sample Characteristics ^a,b^	Inclusion Criteria	Intervention ^c^	Control/Comparison ^c^	Intervention Mode	Intervention Description	Intervention Duration and Contact	Attendance/Attrition or Adherence
Stern et al. (2004), USA Randomised trial	Obesity	*n* = 132 (83% male) Age (y): LC = 53 (9); CD = 54 (9) BMI: LC = 42.9 (6.6); CD = 42.9 (7.7) Weight (kg): LC = 130 (23); CD = 132 (27)	BMI ≥ 35 kg/m^2^	Low-carbohydrate diet [LC], *n* = 64	Conventional (low-fat/low-calorie) diet [CD], *n* = 68	In-person group	Counselling to restrict carbohydrates or calories. LC: ↓ carbohydrate intake to <30 g/day; CD: ↓ caloric intake by 500 calories/day, with <30% calories from fat.	12 months. Weekly counselling sessions (2-h) for 4 weeks, then monthly sessions (1-h) for 11 months.	6-month follow-up: *n* = 79; 1-year: *n* = 87. 6-month weight data carried forward or obtained from medical records = (89%). 1-year weights obtained from medical records = (final 96%)
Boutelle et al. (2005), USA Randomised controlled feasibility trial	Overweight or obese	*n* = 26 (80% male) Age (y): 53.8 (10.4) BMI: 34.7(4.9) Weight (kg): 111.7 (21.3)	≥20% above ideal body weight	Educational mailings re weight management ^d^	Usual care ^d^	Weekly individual mailings and phone calls	Both conditions: At initial visit, weight management discussed. IG: weekly educational mailings weeks 2–7 to address weight management issues (self-monitoring, stimulus management, relapse prevention, ↓ fat content, meal planning, exercise).	8 weeks.	Attendance at 8-week follow-up, IG: 87%; UC: 54.5% (*p* = 0.095).
Yancy Jr et al. (2010), USA Randomised controlled trial	Overweight or obese	*n* = 146 (72% male) Age (y) (SD): LCKD = 52.9 (10.2) OLFD = 52 (9) BMI (SD): LCKD= 39.9 (6.0) OLFD = 38.8 (7) Weight (kg): LCKD = 123.1 (25.4) OLFD = 117.4 (26.0) WC (cm): LCKD = 127.4 (16.9), OLFD = 124.8 (17.4)	BMI between 27 and 30 kg/m^2^ and an obesity-related disease; OR, a BMI of ≥30 kg/m^2^ regardless of comorbidity	Low-carbohydrate, ketogenic diet (LCKD), *n* = 81 allocated (*n* = 72 received)	Orlistat therapy combined with low-fat diet (OLFD), *n* = 79 (*n* = 74 received)	In-person group	Counselling sessions covered topics parallel between the 2 interventions but specific to diet. Advice re exercise, hydration, ↓ of caffeine/alcohol. Restrict carbohydrate intake to <20 g/day.	48 weeks. Meetings every 2 weeks for 24 weeks, then every 4 weeks for 24 weeks. (meeting duration = 1 to 2 h)	Post-intervention numbers, LCKD: *n* = 52; OLFD: *n* = 61. Completion of measurements at 48 weeks: 57 of LCKD group (79%) and 65 of OLFD group (88%). Attendance at ≥80% of group counselling sessions: LCKD *n* = 26; OLFD *n* = 27.
Shahnazari et al. (2013), USA Randomised controlled trial	Overweight or obese	*n*= 50 (84% male) Age (y) [M (95%CI)]: CG = 55 (51, 59), IG = 54 (49, 59) BMI [M (95%CI)]: CG = 30 (28, 32), IG = 31 (29, 33) Weight (kg) [M (95%CI)]: CG = 89 (81, 97), IG = 93 (87, 99)	Veterans responsible for own food selection, preparation, consumption	Individualised wellness coaching, *n* = 43	Initial 1-h nutrition education session only, *n* = 41	In-person individual or via telephone	Nutrition coaching sessions (healthy eating habits, food choices, label reading, cooking techniques, stages of change model to alter eating behaviours). Nutrition education material. Sessions focused on ↓ intake of sugar, salt, high fat meat, fast foods, etc. and ↑ fruit/vegetables, whole-grain, dairy, lean meat, fish, water.	6 months. IG (3.75 h contact): 1× 60 min education session; individualized 15 min wellness coaching weekly in 2nd month; 1× monthly for following 4 months; one final 60 min session.	IG *n* = 28 (65%), CG *n* = 22 (54%) (*p* = 0.284) completed.
Yancy et al. (2015), USA Doubly randomised preference trial	Obesity	*n* = 207 (73% men) Age (y): 55 (11) BMI: 36 (6) Weight (kg): 108 (20) WC (cm): 46 (5)	BMI ≥ 30 kg/m^2^	Choice of diet, *n* = 105, (*n* = 61 low-carb diet [Choice-LCD]; *n* = 44 low-fat reduced calorie diet [Choice-LFD])	No choice, *n* = 102 (*n* = 53 CG-LCD; *n* = 49 CG-LFD)	In-person group and individual telephone	Choice arm: advised if food preferences aligned with LCD or LFD based on food preference questionnaire. CG: Randomised to diet. LCD: carbohydrate intake 20 g/day; calories not restricted. LFD: total fat, saturated fat and cholesterol intake restricted to <30%, <10% and <300 mg/day of daily energy intake; calorie intake restricted by ↓ 500 kcal from daily maintenance energy requirement.	48 weeks. Weeks 1–24: group sessions every 2 weeks. Weeks 25–48: group sessions every 4 weeks, telephone call from dietitian between sessions. 19 visits in total.	Completed intervention: Choice-LCD *n* = 47/61, Choice-LFD *n* = 34/44, CG-LCD *n* = 45/53, CG-LFD *n* = 35/49. Completed weight measurements at 48 weeks, Choice: 87 (83%); CG: 88 (86%). Attendance, number of group sessions attended (of 19) and calls completed (of 6) [m (SD)]: Choice = 13.5 (5.5) and 2.5 (2.5), respectively; CG = 14.8 (4.7) and 3.0 (2.5). Attendance-at least 15 group sessions, Choice: 55.2%; CG: 67.6%. Dietary adherence similar between arms (*p* = 0.66).
Conley et al. (2018), Australia Parallel group randomised controlled trial	Obesity	*n* = 24 (100% male) Age (y): SERD = 67.1 (3.9), IER = 68 (2.7) BMI: SERD = 36.2 (4.3), IER = 33.4 (1.8) Weight (kg): SERD = 107.3 (17.1), IER = 99.1 (7.9) WC (cm): SERD = 122.5 (10.4), IER = 114.2 (5.2)	BMI ≥ 30 kg/m^2^	Intermittent Energy Restriction (IER) 5:2 diet plan, *n* = 11	Standard Energy Restricted Diet (SERD), *n* = 12	In-person individual; telephone (if required)	IER diet: ‘fast’ for 2 non-consecutive days/week (daily calorie intake 600 calories) and eat ad libitum on remaining 5 days. SERD: continuous daily energy-restricted diet (500-calorie daily reduction from average requirement).	6 months. Five individual counselling sessions at weeks 2, 4, 8 and 3 months. Telephone assistance if required from months 4–6.	Post-intervention at 6 months: *n* = 23 (*n* = 1 withdrawal, IER group). Adherence at 3 months: 83% SERD and 82% 5:2 following respective diets. Adherence at 6 months: 75% SERD and 73% IER.
Wu et al. (2007), Taiwan Randomised controlled trial	Overweight or obese inpatients with schizophrenia	*n* = 53 (42% male) Age (y): CRD = 42.4 (7.5), UC = 39.0 (6.7) BMI: CRD = 30.43 (4.2), UC = 30.27 (3.31) Weight (kg): CRD= 78.4 (11.6), UC = 77.8 (112.0) WC (cm): CRD = 98.30 (7.33), UC = 97.82 (9.67) HC (cm): CRD = 108 (8.5), UC = 106.1 (6.5) Waist-to-hip ratio: CRD = 0.91 (0.07), UC = 0.92 (0.07) Body fat %: CRD = 36.9 (7.8), UC= 38 (10.6)	Taking ≥300 mg of oral clozapine per day for at least a year, BMI > 27 kg/m^2^, DSM-IV diagnosis of schizophrenia.	Calorie-restricted diet (CRD) and physical exercise, *n* = 28	Usual care (UC), *n* = 25	Inpatient program	Caloric intake restricted to 1300 to 1500 kcal/day for women; 1600 to 1800 kcal/day for men. Macronutrient intake complied with expected changes of 20%, 25% and 55% in energy from protein, fat, carbohydrate.	6 months.	Post-intervention numbers: *n* = 53 (*n* = 3 withdrew from UC, as discharged from hospital in second month of study).
Niv et al. (2014), USA Pre-post intervention	Obese with schizophrenia or schizoaffective disorder	*n* = 109 (90.8% male) Age (y): 50.2 (9.7) BMI: 31.5 (5.4) Weight (kg): 97.84 (19.78)	Schizophrenia or schizoaffective disorder as per Structured Clinical Interview for DSM-IV diagnosis	Enhancing Quality-of-care In Psychosis (EQUIP): psychosocial weight management program, *n* = 55	Eligible for EQUIP but chose not to enrol, *n* = 54	In-person group; in-person individual as needed	EQUIP Program sessions focused on weight management techniques, light physical exercise. Education on nutritional principles and behavioural techniques to adjust unhealthy eating and exercise habits. Handouts, knowledge quizzes, learning principles adapted for schizophrenia. Nurse care coordinators available as needed.	16 weeks (weekly sessions).	Post-intervention numbers: *n* = 50 (five attended only one session, and thus were dropped from the weight and BMI analysis). Attendance, group sessions: 100% (*n* = 55) attended at least one; average of 3.8 (SD = 4.6) sessions. Individual sessions: 75% (*n* = 41) attended at least one session; average of 3.9 (SD = 3.0) sessions. In total, average of 6.7 (SD = 5.2) in-person sessions (range 1–23).
Iqbal et al. (2010), USA Randomised Controlled Trial	Type II diabetes	*n* = 144 (90% male) Age (y): 59.4 (9.2) BMI: 37.6 (5.5)	Clinically diagnosed Type II diabetes mellitus BMI ≥ 30 kg/m^2^	Low-carb diet (LCD), *n* = 70; or low-fat diet (LFD), *n* = 74	Nil	In-person group	LCD: carbohydrate intake 30 g/day LFD: consume ≤ 30% calories from fat and deficit of 500 kcal/day. To consume <7% total calories from saturated fats; <300 mg of dietary cholesterol daily.	24 months. Weekly 2-h classes for first month; then every 4 weeks.	Post-intervention: *n* = 68; Attrition rate: LCD = 60%, LFD = 46% Attendance: mean 9.9 sessions attended.
North and Palmer (2015), USA Retrospective chart review	Type II diabetes	*n* = 359 medical records (100% male) Age (y): Basics = 65.2 (8.87), CG = 66.8 (9.67)	Diagnosis of Type II DM within previous 2 y.	Diabetes group education (Basics), *n* = 175	Standard diabetes management follow-up (CG), *n* = 184	In-person group sessions	Basics program (3 sessions). Nutrition related topics include: carbohydrate counting, nutrition labels, blood pressure, cholesterol, general healthy eating, weight loss plans.	4 months. 3 sessions: 2.5, 2, and 1.5 hrs duration. Sessions 1 and 2 held 2 weeks apart, Session 3 held 3 months after session 2.	Attendance: each patient in treatment group completed all three sessions of Basics diabetes education program.
Dexter et al. (2019), USA Retrospective chart review	Prediabetes and diabetes	*n* = 75 (88% male) Age bracket (y) *n* (%): 41–50: 8 (10.7%) 51–60: 15 (20.0%) 61–70: 46 (61.3%) 71 +: 6 (8.0%) BMI: face-to-face = 36.6 (6.98), telehealth = 36.32 (4.63)	BMI ≥ 25 and either (a) current Dx of Type II DM or (b) Dx of prediabetes	Healthy teaching kitchen (HTK), *n* = 75	Nil	In-person group or via Clinical Video Telehealth	Multicomponent intervention. Cooking and nutrition education class topics included carbohydrate counting, meal planning, creating recipes.	12 weeks. HTK offered once every 12 weeks.	NA
Friedberg et al. (2015), USA Randomised controlled trial	Uncontrolled hypertension	*n* = 533 (99% male) Age (y) [M (SE)]: SMI = 66.4 (0.66), HEI = 66.5 (0.96), UC = 65.4 (0.76) BMI [M(SE)]: SMI = 30.5 (0.38), HEI = 31.2 (0.47), UC = 30.0 (0.34) BP control (%): SMI = 42.6, HEI = 40.6, UC = 44.6 (*p* = 0.74) SBP (mmHg) [m(SE)]: SMI = 136.0 (0.89), HEI = 137.2 (1.33), UC = 137.0 (0.96) (*p* = 0.65)	Hypertension, antihypertensive drug therapy for ≥6 months, uncontrolled BP during screening.	Hypertension diet. Stage-matched intervention (SMI), *n* = 176	Health Education Intervention (HEI), *n* = 177; or Usual Care (UC), *n* = 180	Individual telephone counselling	SMI: tailored monthly phone counselling for exercise, diet and medications based on current stage of change. HEI: monthly telephone counselling of non-tailored information on HT, diet, medication, exercise guidelines.	6 months. SMI: monthly phone counselling (approx. 30 min). HEI: monthly telephone counselling (approx. 15 min)	6 months: *n* = 481, (6-month missing data rate of <10%).
Sikand et al. (1998), USA Retrospective chart review, observational cohort	Primary hypercholesterolaemia	*n* = 95 (100% male) Age (y): 60.9 (9.9) BMI: 26.9 (0.5)	Primary Dx hypercholesterolemia, previously met NCEP criteria for initiating cholesterol-lowering drug therapy, not on cholesterol-lowering medication, aged 21–75 y.	National Cholesterol Education Program (NCEP) dietary intervention phase of a clinical trial, *n* = 95	Nil	In-person individual	Dietitian initiated “medical nutrition therapy” to ↓ elevated cholesterol levels, progressively ↓ saturated fat and cholesterol intake, promote weight loss by eliminating excess total calories. I.e. intake of total fat <30% of calories, saturated fatty acids <10% of calories and cholesterol <300 mg/day.	8 weeks. 2–4 visits (120–180 min).	Post-intervention: *n* = 74 (78% complete data). Attendance: mean no. of dietitian visits = 2.8 (0.7) (range = two to four visits); mean total time = 144 (21) minutes over 6.8 (0.7) weeks. Two visits (*n* = 29); three visits (*n* = 33); and four visits (*n* = 12).
Tan-Shalaby et al. (2016), USA Observational safety and feasibility trial (pre-post)	Advanced malignant cancers	*n* = 17 (100% male) Age (y): 65 (11.7) BMI: 29.46 (5) Weight (kg): 92 (2.3)	Advanced solid malignancies measurable on FDG PET/CT imaging; not on chemotherapy.	Modified Atkins Diet (Ketogenic diet) *n* = 17	Nil	In-person individual	20 to 40 g of carbohydrates/day, advised on grocery shopping, menu planning. Restricted consumption of high carbohydrate foods. No restriction on calories, protein, fats.	16 weeks.	Screen failures in 6/17 (35%). Adherence: 11/17 proceeded with trial. 6 (35%) maintained diet for 8 weeks, 4 (23%) completed 16 wks. Three successfully dieted >16 wks. One died at 80 weeks and one at 116 weeks. One alive without evidence of disease at 131 wks.
Holton et al. (2020), USA Cohort study	Gulf War Illness (GWI)	*n* = 46 (72.5% male) Age (y): 54.35 (6.02) BMI: 32.10 (5.34)	Active deployment during Gulf War; symptoms meeting Kansas criteria and CDC criteria for GWI.	Low Glutamate Diet, *n* = 20	Wait-listed control group (started diet 1 month later), *n* = 20	Individual via Skype	Whole food diet, restricting free glutamate and aspartate. Provision of materials including list of foods to avoid, food additives to avoid, high antioxidant food list, shopping list, sample recipes.	1 month.	Post-treatment: *n* = 40; Adherence: *n* = 34 at 3 months post-completion, reported 30/34 (88%) were still following the diet at this time point.
Bayer-Carter et al. (2011), USA Randomised controlled trial	Amnestic mild cognitive impairment (aMCI) vs. healthy controls (HC)	*n* = 49 (47% male) Age (y): HC = 69.3 (7.4), aMCI = 67.6 (6.8) BMI [M (SEM)]: HC (LOW) = 26.4 (2.6), HC (HIGH)= 29.5 (4.5), aMCI (LOW) = 27.4 (3.8), aMCI (HIGH) = 27.5 (3.4)	aMCI diagnosed as delayed memory scores deviating ≥1.5 SD from estimated premorbid ability	LOW Diet (low-saturated fat/low glycaemic index diet), *n* = 25	HIGH diet (high-saturated fat/high-glycaemic index), *n* = 24	Food delivered to participants’ homes twice weekly	The HIGH diet (fat, 45% [saturated fat, >25%]; carbohydrates, 35%–40% [glycaemic index, >70]; and protein, 15%–20%). LOW diet (fat, 25%; [saturated fat, <7%]; carbohydrates, 55%–60% [glycaemic index, <5]; and protein, 15%–20%).	4 weeks.	Diet adherence: Mean non-adherent incidents per week ranged from 1.23 to 1.80 per group.
Serra et al. (2021), USA Single group prospective feasibility trial	Medically stable nursing home residents	*n* = 50 (92% Male) Age (y) (M ± SEM): 69.7 ± 0.7 BMI (M ± SEM): 31.7 ± 0.9	Medically stable > 64 y; enrolled in Gerofit clinical demonstration program.	Nutrition education	Nil	In-person group and optional individual counselling	Registered Dietitian (RD) led classes addressing age-related nutrition concerns: nutrition basics, food labels, hydration, meal planning, the DASH diet, protein intake, food shopping on budget. Individualized nutrition counselling sessions available.	7 months. Class duration: 30 min.	Attended ≥ 1 group class = 39 (78%) Attended ≥ 2 group classes = 24 (62%) (M ± SEM: 2.9 ± 2.0 classes, which was 82% of total available sessions).

Abbreviations: BP—blood pressure; BMI—Body Mass Index; CG—control/comparison group; DASH—Dietary Approaches to Stop Hypertension; DM—Diabetes Mellitus; DSM-IV—Diagnostic and Statistical Manual of Mental Disorder, 4th edition; FDG PET/CT—fluorodeoxyglucose positron emission tomography/computed tomography; HC—Hip Circumference/Healthy Control; IER—Intermittent Energy Restriction; IG—intervention group; HEI—Health Education Intervention; HT—hypertension; LCD—Low Carbohydrate Diet; LFD—Low Fat Diet; NCEP—National Cholesterol Education Program; SBP—systolic blood pressure; SERD—Standard Energy Restricted Diet; SMI—Stage Matched Intervention; UC—Usual care; WC—Waist circumference; (y) —Years. ^a^ *n* = refers to total sample size; data are presented as mean (SD) unless otherwise stated; ^b^ BMI units = kg/m^2^; ^c^ *n* = refers to number of individuals per group; ^d^ number of individuals per group not reported for this study.

**Table 4 nutrients-14-02094-t004:** Health-related outcome measures of the included studies, grouped and listed in order of health condition discussed in the text.

Study	Dietary Intervention	Health-Related Outcome Measures	Within Group Comparisons ^a,b^	Between Group Comparisons ^a,b^
Stern et al., (2004)	Low-carbohydrate diet	Weight; blood lipids (TC, LDL-C, HDL-C, triglycerides); HbA1c; SBP and DBP; insulin level; glucose level		**At 1 year** **Mean change (SD); Mean difference (95% CI)** **Weight (kg):** LCD = −5.1 (8.7), CD = −3.1 (8.4); −2.0 (−4.9, 1.0) (*p* = 0.195) **TC:** LCD = 0.16 (1.11), CD = −0.21 (0.91) (*p* = 0.143) **LDL-C:** LCD = 0.18 (0.91), CD = −0.10 (0.75) (*p* = 0.191) **HDL-C (mmol/L):** LCD = −0.03 (0.18), CD = −0.13 (0.16); 0.08 (0.01, 0.16) (*p* = 0.028) **Triglycerides (mmol/L):** LC = −0.65 (1.78), CD = 0.05 (0.96); −0.62 (−1.09, −0.15) (*p* = 0.044) **HbA1C (participants with diabetes, *n* = 54), %:** LCD = −0.7 (1.0), CD = −0.1 (1.6); −0.7 (−1.6, 0.2) (*p* = 0.102) **SBP (mmHg):** LC = 1 (19), CD = 2 (15) (*p* = 0.78) **DBP (mmHg):** LC = 3 (15), CD = 1 (10) (*p* = 0.502) **Insulin level (participants with diabetes):** LC = −35 (236), CD = −28 (139) (*p* = 0.917) **Glucose level (participants with diabetes) mm/L:** LC = −1.55 (2.16), CD:−1.17 (3.66) (*p* = 0.800)
Boutelle et al., (2005)	Educational mailings	Weight; Food habits (FHQ); Health status (MOS SF-36); readiness to change (URICA)		**At 8 weeks:** **Weight (lbs):** IG = −3.91 (6.15); CG = −1.62 (3.33), *p* = 0.138 **Weight loss (% participants):** IG = 67%, CG = 36% (*p* = 0.129) **FHQ (Fruit and vegetable consumption subscale):** Group X Time interaction *p* = 0.019; IG mean change (SD) = −0.56 (0.58), *p* = 0.004 (↑consumption). CG = 0.22 (0.69), *p* = 0.47 (↓consumption). **FHQ, other subscales:** ns **URICA:** All subscales ns
Yancy Jr et al., (2010)	Low-carbohydrate, ketogenic diet (LCKD)	Weight; WC; BP; blood lipids (TC, triglycerides, LDL-C, HDL-C); metabolic indices (fasting glucose level, fasting insulin level, HbA1c)	**At 48 weeks:** Mean change (95%CI) **Weight:** LCKD = −11.4 kg (−14.8, −7.9), % loss: −9.5% (−12.1, −6.9); OLFD = −9.6 kg (−11.9, −7.3), % loss: −8.5% (−11.0, −6.1); **WC:** LCKD = −11.07 cm (−13.85, −8.29); OLFD = −11.07 cm (−13.85 to −8.29); **SBP:** LCKD = −5.94 (−8.80, −3.08); OLFD = ns **DBP:** LCKD = −4.53 (−6.57, −2.49); OLFD = ns **TC:** LCKD = ns; OLFD = −8.86 mg/dL (−15.31 to −2.41) **Triglycerides:** LCKD = −28.83 mg/dL (−48.08, −9.58); OLFD = −21.40 mg/dL (−39.63, −3.17) **LDL-C:** LCKD = ns; OLFD = −8.29 (−14.06, −2.52) **HDL-C: **LCKD = ns; OLFD= 3.42 (1.61, 5.25) **TC/HDL-C:** LCKD = −0.44 (−0.68, −0.20); OLFD = −0.51 (−0.73, −0.28) **Triglyceride/HDL-C:** LCKD = −1.00 (−1.65, −0.35); OLFD = −0.77 (−1.38, −0.15) **Fasting glucose level:** LCKD = −9.7 mg/dL (−16.9, −2.6); OLFD = −3.26 mg/dL (−10.05, −3.54) **Fasting insulin level (participants without diabetes only):** LCKD = −7.3 IU/mL (−13.5, −1.2); OLFD = ns **HbA1c:** LCKD = −0.3% (−0.5, −0.1); OLFD = ns	**At 48 weeks:** Mean difference (95% CI) LCKD and OLFD **Weight (%):** −0.95 (−4.50, 2.61) (*p* = 0.60) **Weight (kg):** −1.75 (−5.90, 2.40) (*p* = 0.41) **WC (cm):** ns **SBP (mm Hg):** −7.44 (−11.12, −3.75) (*p* = 0.001) **DBP (mm Hg):** −4.97 (−7.64, −2.29) (*p* = 0.001) **Blood lipids and metabolic indices:** ns
Shahnazari et al., (2013)	Individualised wellness coaching	Weight; BMI; Block 2005 Food Questionnaire; readiness to improve eating behaviour (SOCMII)	**Pre (baseline) vs. post (6 months):**^d^ **Weight (kg):** IG = 92.8 (4.1) vs. 88.2 (3.4) (*p* = 0.006), CG = 88.7 (5.4) vs. 87.4 (4.6) (*p* = 0.33) **BMI ^e^:** IG = 30.4 vs. 28.9 (*p* = 0.02), CG = 29.6 vs. 29.1 (*p* = 0.37) **Nutrients and Intakes:** Refer to paper for full details *. **SOCMII:** IG = 0.32 (0.09) vs. 0.8 (0.15) (*p* = 0.006); CG = ns	
Yancy et al., (2015)	Choice of diet: low-carbohydrate diet, or low-fat reduced calorie diet (Choice)	Weight; WC; dietary adherence (FFQ); QoL (IWQOL-Lite)	**At 48 weeks:** **Weight (kg) [M(95% CI)]:** Choice = −5.7 (4.3, 7.0); CG = −6.7 (5.4, 8.0)	**At 48 weeks:** Estimated mean difference, Choice-CG [95% CI] **Weight (kg):** 1.1 (−2.9, 0.8); *p* = 0.26 **WC (cm):** 0.4 (−0.3, 1.3); *p* = 0.28 **Dietary adherence (FFQ):** −0.9 [−4.9,3.1] *p* = 0.66 **IWQOL-Lite:** −0.8 (−4.1, 2.6) *p* = 0.65
Conley et al., (2018)	Intermittent Energy Restriction (IER) diet plan (The 5:2 Diet)	Weight; BMI; WC; BP; FBG; blood lipids (TC, LDL-C, HDL-C, triglycerides); FFQ (energy, protein, fat, CHO, sugars, fibre, calcium, alcohol, sodium); QoL (AQoL-8D)	**At 6 months:** **Weight (kg):** IER = −5.3 (3.0) (*p* < 0.001), SERD = −5.5 (4.3) (*p* < 0.001) **BMI:** ↓ in both groups approx. −2.2 (*p* < 0.001). SERD = 34.4 (5.3), IER = 31.5 (2.2) **WC (cm):** SERD = -−6.4 (10) (*p* < 0.001), IER = −8 (10) (*p* < 0.001) **SBP (mmHg):** SERD = −10.2, IER = −14 (*p* < 0.001) **DBP (mmHg):** *p* = 0.2 **FBG:** *p* = 0.15 **TC:** *p* = 0.52 **LDL-C:** *p* = 0.6 **HDL-C:** *p* = 0.68 **Triglycerides:** *p* = 0.22 **QoL ^c^: ** Both groups improved psycho-social dimension score, (*p* = 0.001); physical dimension score (*p* = 0.03); and overall AQol-8D score (*p* = 0.003) **FFQ:** Significant ↓ in all nutrient intakes (except alcohol, ns) in both groups	**At 6 months ^c^:** **Weight (kg):***p* = 0.79 **BMI:** *p* = 0.85 **WC (cm):** *p* = 0.54 **SBP (mmHg):** *p* = 0.76 **DBP (mmHg): ***p* = 0.83 **FBG:** *p* = 0.81 **TC:** *p* = 0.69 **LDL-C:** *p* = 0.86 **HDL-C:** *p* = 0.93 **Triglycerides:** *p* = 0.78 **QoL ^c^:** N/A **FFQ:** All ns change between groups
Wu et al., (2007)	Calorie-restricted diet (CRD) and physical exercise	Weight; BMI; body fat %, waist to hip ratio; blood lipids and metabolic parameters (serum glucose, triglyceride, cholesterol, insulin)	**At 6 months:** **Weight (kg):** CRD = –4.2 (4.4) (*p* < 0.001), CG = 1.0 (3.4) ns **BMI:** CRD = –1.59 (1.66) (*p* < 0.001), CG = 0.35 (1.30) ns **WC (cm):** CRD= 3.32 (4.18) (*p* < 0.001), CG = 1.02 (4.25) ns **HC (cm):** CRD = –3.3 (4.5) (*p* < 0.001), CG = 0.3 (2.7) ns **Body fat %:** CRD = –1.3 (6.4) ns, CG = 1.3 (4.2) ns **Waist to hip ratio:** CRD = −0.01 (0.4) ns, CG = 0.01 (0.3) ns **At 6 months (*p* value for baseline to 6mth):** **Glucose (mg/dL):** CRD = 96.4 (16.9) ns, CG = 99.8 (16.9) ns **Triglyceride (mg/dL):** CRD = 146.8 (90.9) (*p* < 0.001), CG = 239.3 (188.9) ns **Cholesterol (mg/dL):** CRD = 159.2 (36.9) ns, CG = 166.8 (35.2) ns **Insulin (µIU/mL):** CRD = 5.2 (3.1) (*p* < 0.001), CG = 9.5 (9.2) ns	**At 6 months, CRD vs. Control:** **Weight (kg):***p* < 0.001 **BMI:** *p* < 0.001 **WC (cm):** *p* < 0.001 **HC (cm):** *p* < 0.05 **Body fat %:** ns **Waist to hip ratio:** ns **At 6 months:** **Glucose (mg/dL):** ns **Triglyceride (mg/dL):** *p* < 0.05 **Cholesterol (mg/dL):** ns **Insulin (µIU/mL):** ns
Niv et al., (2014)	Enhancing Quality-of-care In Psychosis (EQUIP)-psychosocial weight management program	Weight; BMI	**At week 16, EQUIP:** **Weight (lbs):**−2.4 (10.6), (t = 1.6, *p* = 0.12) **BMI:** −0.3 (1.5), (t = 1.5, *p* = 0.13) **At 12 months: Weight:** significant ↓in weight in both groups over time [F(1, 94) = 5.2, *p* < 0.05] **BMI:** significant ↓ in BMI in both groups over time [F(1, 94) = 5.7, *p* < 0.05]	**At 12 months:** **Weight (lbs):** EQUIP = −2.3 (18.0), CG = −2.2 (11.9); time x treatment group [F(1, 94) = 1.2, *p* > 0.05] **BMI:** EQUIP = −0.3 (2.6); CG = −0.3 (1.7); time x treatment group [F(1, 94) = 1.3, *p* > 0.05]
Iqbal et al., (2010)	Low-carbohydrate diet (LCD) or low-fat diet (LFD)	Weight; glucose; HbA1c; serum lipids; BP; dietary intake (24 hr recall)		**At 24 months: (ITT analysis)** **Weight (kg):** LC = −1.5, LF = −0.2; **Glucose (mg/dL):** ns **HbA1c:** ns **All Serum lipids:** ns **SBP and DBP (mmHg):** ns **Dietary intake:** ns
North and Palmer, (2015)	Diabetes education	HbA1c; weight; SBP		**Post Hoc (4-months):** **HbA1c ^c^:** IG = 7.18 (1.19), 6.61 (0.80); CG = 6.68 (0.61), 6.69 (0.74) (*p* < 0.001) **Weight (lbs):** IG = 228.95 (43.63), 224.33 (42.66); CG = 231.39 (45.27), 229.72 (45.27) (*p* < 0.001) **SBP ^c^:** IG = 131.83 (17.43), 126.01 (15.84); CG = 131.33 (15.04), 128.01 (13.07) ns
Dexter et al., (2019)	Healthy teaching kitchen	Weight; BMI; metabolic parameters (HbA1c, TC, LDL-C, HDL-C, Triglycerides); Questionnaires: cooking frequency (0–3), cooking confidence (0–5), fruit and vegetable incorporation (0–5), confidence in healthy cooking (0–5); Healthy Habits Questionnaire (HHQ, 0–75)	**At 12 weeks:** Mean change (SD) **BMI:** −0.35 (0.76) (*p* < 0.05) **Weight (lbs):** −2.91 (5.75) (*p* < 0.05) **Metabolic parameters:** ns **HbA1c:** −0.13 (0.94) ns **TC:** 0.63 (25.75) ns **LDL-C:** 0.57 (19.22) ns **HDL-C:** 0.93 (5.62) ns **Triglycerides:** −18.56 (113.11) ns **Cooking confidence:** 0.55 (1.07) (*p*< 0.001) **Confidence in incorporating fruits and vegetables:** 0.83 (0.89) (*p* < 0.001) **Confidence in preparing healthy meals:** 0.62 (1.01) (*p* < 0.001) **Cooking frequency:** 0.16 (0.70) ns **HHQ (% change):** Improvement in 9/10 questionnaire responses (*p* < 0.05)	
Friedberg et al., (2015)	Hypertension diet. Stage-matched intervention (SMI) or health education intervention (HEI)	BP; Diet adherence (DASH)	**At 6 months:** **Improvement in proportion of participants with controlled BP:** SMI = 19.7% (*p* < 0.001), HEI = 11.9% (*p* = 0.012), UC = 1.3% (*p* = 0.76)	**At 6 months:** **Participants with controlled BP (%):** SMI = 64.6, HEI = 54.3, UC = 45.8; (SMI vs. UC *p* = 0.001; HEI vs. UC *p* = 0.108) **SBP [M(95% CI)] (mmHg):** SMI = 131.2 (129.1, 133.3), HEI = 131.8 (129.9, 133.7), UC = 134.7 (132.7, 136.7); (SMI vs. UC *p* = 0.009; HEI vs. UC *p* = 0.047) (ns when adjusting for multiple comparisons) **At 6 months:** **Change in SBP [M (95% CI)]:** SMI = −4.7 (−6.9, −2.5), HEI = −5.4 (−8.5, −2.3), UC = −2.7 (−5, −4); (SMI vs. UC *p* = 0.007; HEI vs. UC *p* = 0.009) **Change in DASH score [M (95% CI)]:** SMI = 0.69 (−0.1, 1.5), HEI = −0.16 (−1.1, 0.8), UC = −0.76 (−1.5, 0) (SMI vs. UC *p* = 0.01; HEI vs. UC *p* = 0.32)
Sikand et al., (1998)	National Cholesterol Education Program (NCEP) Step 1 dietary intervention phase	Metabolic parameters (TC, LDL, HDL, triglycerides); BMI	**At 8 weeks:** % change, mean actual change value (SE) **TC(mmol/L):** −13.4%; −0.94 (0.09) (*p* < 0.0001) **LDL (mmol/L):** −14.2%; −0.76 (0.08) (*p* < 0.0001) **HDL (mmol/L):** −4.4%; −0.05 (0.02) (*p* < 0.05) **Triglyceride:** −10.8%; −0.22 (0.87) (*p* < 0.05) **TC/HDL:** −9.4%; −0.59 (0.14) (*p* < 0.001) **LDL/HDL:** −12.1%; −0.54 (0.11) (*p* < 0.001) **BMI:** 0.4%; −0.1 (0.52) ns	
Tan-Shalaby et al., (2016)	Modified Atkins Diet (Ketogenic diet)	Safety and feasibility: QoL (EORTC QLQ-c30); Weight; BMI; BP; blood lipids (TC, HDL-C, LDL-C, triglycerides); fasting glucose	**Baseline to last visit:** **Weight (kg):** baseline = 95 (18.7), last visit = 87.7 (37.82); change = −7.5 (5.8) (*p* < 0.0001) **BMI:** −2.669 (1.99) (*p* < 0.001) **Other biochemical parameters:** ns **DBP:** baseline = 70 (11.27), last visit = 77 (5.57) (*p* = 0.043) **QoL, pts on diet for 4+ weeks (*n* = 6):** ns	
Holton et al., (2020)	Low Glutamate Diet	Total symptom score (TSS, 0–33); Improvement (≥30% symptom remission); Patient Global Impression of Change Scale (PGIC); Chalder Fatigue Scale	**At 1 month:** **TSS:** pre-diet = 21 (5), post-diet = 12 (5) (*p* < 0.0001) **Improvement:** IG = 65% participants **PGIC:** IG = 73% improved **Chalder Fatigue Scale:** pre-diet = 29 (8), post-diet = 16 (9) (*p* < 0.0001)	**At 1 month:** **TSS:** IG = 11.7 (5.3), CG = 18.1 (5.7) (*p* = 0.0009); effect size d = 1.16
Bayer-Carter et al., (2011)	HIGH (high-saturated fat/high-glycaemic index diet) or LOW (low-saturated/low-glycaemic index diet)	Insulin and glucose levels; Homeostasis Model Assessment of Insulin Resistance (HOMA-IR); blood lipids (TC, LDL-C, HDL-C); weight	**At week 4 ^c^:** **Insulin AUC:** For both diagnoses, HIGH diet ↑ and LOW diet ↓ insulin AUC (time x diet *p* = 0.01) **Glucose AUC:** ns **HOMA-IR:** ns (time x diet *p* = 0.06) **Weight:** ns **TC:** differed between groups (time × diet × diagnosis *p* = 0.04)-due to HIGH diet ↑ TC and LOW diet ↓ TC for both diagnoses (time × diet, *p* < 0.001). 2-fold greater changes in aMCI compared to healthy participants. Change scores for both diagnostic groups in either diet condition = ns **LDL:** time × diet × diagnosis *p* = 0.048; time × diet *p* < 0.001; no differences between CG vs. aMCI diagnoses change scores. **HDL:** ↑ with HIGH diet and ↓ with LOW diet (time × diet *p* < 0.001) **LDL/HDL:** ↑ by HIGH diet and ↓ by LOW diet for both diagnoses (time × diet *p* = 0.04)	
Serra et al., (2021)	Nutrition education	Behavioural Risk Factor Surveillance System (BRFSS); F&V intake Questionnaire, self-rated diet quality (VAS)	**Baseline to last class:** In those attending ≥ 2 group classes (*n* = 24): **Daily F&V intake [M(SEM)]:** baseline = 3.4 (1.9), post-intervention = 4.1 (2.0) servings/day (*p* = 0.07) **Consumption of ≥5 F&V servings/day:** baseline = 21%, post-intervention = 33% (*p* < 0.01) **Self-rated diet quality [M(SEM)]:** baseline = 4.7 (0.5), post-intervention = 5.9 (0.4) (*p* = 0.03)	

Abbreviations: AQoL−8D—Assessment of Quality of Life-8D measure; AUC—Area Under the Curve; BMI—Body Mass Index; BP—blood pressure; CD—conventional diet; CHO—Carbohydrates; CG—Comparison/control group; CRD—Calorie Restricted Diet; DASH—Dietary Approaches to Stop Hypertension; DBP—diastolic blood pressure; EORTC QLQ-c30—the European Organisation for Research and Treatment of Cancer Quality of Life Questionnaire c30; FBG—Fasting Blood Glucose; FFQ—Food Frequency Questionnaire; FHQ—Food Habits Questionnaire; Hba1c—haemoglobin A1c; HDL-C—high-density lipoprotein-cholesterol; IER—Intermittent Energy Restriction diet; IG—Intervention Group; IWQOL-Lite—Impact of Weight on Quality of Life-Lite questionnaire; LCD—low-carbohydrate diet; LCKD—Low Calorie Ketogenic Diet; LDL-C—low-density lipoprotein-cholesterol; LFD—Low Fat Diet; MOS SF-36—Medical Outcomes Study Survey Short Form 36; NR—not reported; ns—not significant; OLFD—Orlistat + Low Fat Diet; QoL—Quality of Life; SBP—systolic blood pressure; SERD—Standard Energy Restricted Diet; SOCMII—Stages of Change Modified Motivational Interviewing tool; TC—total cholesterol; TSS—Total Symptom Score; URICA—The University of Rhode Island Change Assessment Scale; VAS—visual analogue scale; WC—Waist Circumference. ^a^ Results reported as, the mean (SD)—unless otherwise specified; ^b^ BMI units (kg/m^2^) Not reported throughout table; ^c^ values and/or units not reported in paper; ^d^ results assumed to be reported as mean (SE); ^e^ Variance not reported for BMI; * too comprehensive for table.

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
