# Peer review of "Scope of Use and Effectiveness of Dietary Interventions for Improving Health-Related Outcomes in Veterans: A Systematic Review"

_nutrients, 2022, doi:10.3390/nu14102094_

Round 1

Reviewer 1 Report

My comments are:

The research question of this systematic review is interesting. I have the following comments:

The title could be modified and I propose  "Effectiveness of dietary interventions on health outcomes in veterans: a systematic review of randomized controlled clinical trials"

I don't agree to include lifestyle interventions that cannot discriminate the effect of diet from other lifestyle components. 

Randomized controlled trials should be included as they provide higher scientific evidence than single arm or non randomized controlled trials.

For the risk of bias I would suggest to use the second version of the Cochrane Risk of Bias instrument covering the following domains: randomization process, deviation from intended interventions, missing outcomes data, measurement of the outcome, selective outcome reporting, overall bias. 

The data synthesis needs more description.

Since the majority of studies were conducted in subjects with overweight/obesity who lost weight after the interventions meta-analysis would be performed.

Reviewer 2 Report

Mellor et al. submitted a manuscript titled: “Scope of use and effectiveness of dietary interventions for improving health-related outcomes in veterans: a systematic review.”

Dear Author,

Congratulations on your paper. It is an interesting and well written study. However, a minor revision is needed.

Introduction: What is the background of your study? Why was this specific group researched? I would like to see more explanations and clarifications about the reasons you chose this certain population to investigate.

Discussion: In this section, I would like to see more about your opinions and elaborations of the presented results. You should discuss why some of the included studies have poorer outcomes and what could possibly be done in future studies.

Reviewer 3 Report

Comments:

Review of the paper

nutrients-1686237

This paper reviewed the scope of health conditions targeted with dietary interventions and the effectiveness of these interventions for improving health related outcomes in veterans. Although many literatures have mentioned the changes of health status of veterans after retirement and the use of dietary interventions to improve the health of veterans, this paper described these two aspects in more detail and considered the health problems of many veterans, and the impact of interventions on the health profile of veterans in more aspects. The topic of the present study is very interesting. The research is innovative. The manuscript is well written, and the literature results are well discussed. The content of this paper is comprehensive, thus this paper can be accepted and published after several issues are solved.

ABSTRACT

- Line 19:  citations or references or records? Please confirm.

- Line 21:  what is MOVE!? Please add full name.

METHODS

- Is the word “Materials” appropriate? Please refer to the Author Guide.

RESULTS

- How do the authors determine the order of the studies listed in Table 3-6? The publication year, the importance, or the relevance?

Round 2

Reviewer 1 Report

Point 1: The title could be modified and I propose "Effectiveness of dietary interventions on health outcomes in veterans: a systematic review of randomized controlled clinical trials"

Response 1: Thank you for your suggestion. However, the suggested modification is not an accurate depiction of this work, as the review did not exclusively include randomized controlled clinical trials. As specified in the Study Eligibilty Criteria / Study Design section, all studies involving a dietary intervention (randomised controlled trials, non-randomised controlled trials, pseudo-randomised controlled trials, case control studies, single arm pre-test/post-test studies, prospective cohort studies) were eligible for inclusion. Furthermore, this review focused on the scope of use of dietary intervention studies in veterans in addition to effectiveness, which has been included in our original title.

Comment: The reason I suggested to modify the title and add randomized controlled clinical trials is due to the fact that when talking about “Effectiveness of dietary interventions” someone would expect to include studies that are designed as RCTs and those that evaluated the effectiveness of dietary interventions on health outcomes. I do not agree to include non-randomised controlled trials, pseudo-randomised controlled trials, case control studies, single arm pre-test/post-test studies, prospective cohort studies and this is my major concern for the present systematic review.

Point 2: I don't agree to include lifestyle interventions that cannot discriminate the effect of diet from other lifestyle components. 

Response 2: We agree that the effect of a specific diet cannot be discriminated from the outcomes of a multifactorial lifestyle intervention, due to the multiple contributing factors from other lifestyle components included, such as physical activity or psychological support.

However, we included dietary interventions which were part of multifactorial lifestyle or behavioural interventions as this is commonly how numerous medical and psychological conditions are addressed – a holistic, multidisciplinary approach- and it aligns with clinical guidelines. Part of our aim was to determine the scope of dietary interventions which have been investigated across specific veteran populations and including both diet-only and diet as part of combined interventions allowed us to explore this full scope and include pragmatic studies.

Comment: You may add in the limitations part that the components of interventions differed among the included studies increasing the heterogeneity across the included studies and that the effect of diet from other lifestyle components cannot be discriminated.

Point 3: Randomized controlled trials should be included as they provide higher scientific evidence than single arm or non randomized controlled trials.

Response 3: Randomised controlled trials (RCTs) were included in this systematic review. Of the 17 studies that reported on dietary interventions outside of the MOVE! program, 10 represented RCTs. We wonder if the reviewer means that ONLY randomised controlled trials should be included?

We recognise that RCTs are superior to many other study designs in terms of level of evidence. However, the purpose of this systematic review was, firstly, to evaluate the scope of health conditions targeted with dietary interventions and the effectiveness of these interventions for improving health-related outcomes in veterans. If only RCTs were included, the limited number of these types of studies would not be able to address the research question. Hence, for appropriate context, the inclusion of all study designs involving a dietary intervention (randomised controlled trials, non-randomised controlled trials, pseudo-randomised controlled trials, case control studies, single arm pre-test/post-test studies, prospective cohort studies) was decided upon.

Comment: Randomized controlled trials should be included to determine the effectiveness and safety of different dietary interventions on health outcomes in veterans and support the development of interventions to improve dietary behavior at this life-stage to promote health.

Point 6: Since the majority of studies were conducted in subjects with overweight/obesity who lost weight after the interventions meta-analysis would be performed.

Response 6:

For reasons stated above, meta-analysis was not suitable for reporting outcomes of this review. In the studies not associated with the MOVE! Program (which were the primary focus of the results and discussion), the cohorts consisted of a range of populations who were overweight/obese, had type II diabetes, uncontrolled hypertension, advanced cancers, schizophrenia, Gulf War Illness, and cognitive impairment. Even in the studies involving subjects who were overweight/obese, the interventions were varied, with regards to the specific dietary modifications targeted, and primary outcome measures were diverse. These factors all preclude true meta-analysis of these studies. 

Comment: I agree with this response. I suggest to add in the limitations part a sentence highlighting it. For example, say the included studies varied significantly in the type of dietary intervention, and the outcome measurements so that it was difficult to perform a meta-analysis.
